# Cerebral Vascularization and the Remaining Area Supply of the Internal Carotid Artery Derivatives of the Red Kangaroo (*Osphranter rufus*)

**DOI:** 10.3390/ani13172744

**Published:** 2023-08-29

**Authors:** Maciej Zdun, Jakub Jędrzej Ruszkowski, Maciej Gogulski

**Affiliations:** 1Department of Animal Anatomy, Poznan University of Life Sciences, Wojska Polskiego 71C, 60-625 Poznan, Poland; maciej.zdun@up.poznan.pl; 2Department of Basic and Preclinical Sciences, Nicolaus Copernicus University in Torun, Lwowska 1, 87-100 Torun, Poland; 3University Centre for Veterinary Medicine, Szydłowska 43, 60-656 Poznan, Poland; maciej.gogulski@up.poznan.pl; 4Department of Preclinical Sciences and Infectious Diseases, Poznan University of Life Sciences, Wołynska 35, 60-637 Poznan, Poland

**Keywords:** angiology, circle of Willis, *Macropus rufus*

## Abstract

**Simple Summary:**

The red kangaroo (*Osphranter rufus*) is the largest marsupial native to Australia. Its detailed vascular anatomy has not been researched before. Anatomical studies on such species may contribute as a base for physiological studies as well as serve veterinarians in developing diagnostic and treatment protocols for kangaroos in zoos and wildlife rehabilitation centers. In this study, we describe the detailed course of arteries supplying blood to the brain and nearby regions.

**Abstract:**

The red kangaroo (*Osphranter rufus*) is a member of Macropidideae superfamily. It is one of the four kangaroo species living nowadays, and it is the biggest one. It is native to Australia, where it is an abundant species living across the whole continent in stable populations. Outside its natural habit, the red kangaroo is a common species found in zoos and as patients in wildlife rehabilitation centers. Reports on kangaroo anatomy are scarce. Describing detailed anatomy is a base for establishing diagnostic and treatment protocols for different species of animals. Cardiovascular diseases and pathological changes suggestive of hypertension have been previously described in kangaroos. This creates a necessity for detailed studies on species’ vascular anatomy. New reports in the field of detailed vascular anatomy can bring considerable information that complements numerous studies on the evolution or biology of individual species. In this article, we describe the arterial vascularization of the brain and nearby regions of the cranial cavity using various anatomical techniques. The vascularization of the brain is discussed and compared with different mammalian species.

## 1. Introduction

The red kangaroo (*Osphranter rufus*) is a member of the Macropodoidea superfamily. This taxon is a diversified group containing kangaroos, wallabies, and rat kangaroos. The species is endemic to Australia and is widely distributed across the continent. It occurs in arid and semiarid areas [1]. In their natural habitat, red kangaroos are the main large terrestrial herbivores and one of the biggest marsupials living nowadays. Females reach 74.5–110 cm of body length (without the tail) and males 93.5–140. Females can weigh up to 39 kg and males up to 92 kg [2]. Red kangaroos are mostly nocturnal or crepuscular. They become active in the late afternoon or evening and feed on vegetation during the night [1]. It is an abundant species that is a subject of the Management Plan for the Commercial Harvest of Kangaroos in Western Australia provided by the country’s government.

Outside of its natural habitat, the red kangaroo is a common animal kept in zoos around the world. It may also be kept in wildlife rehabilitation centers. Animals kept in both these places are provided with constant veterinary care. With the increase in the number of similar institutions, the medicine for this species is developing. The description of the detailed anatomy of individual species is the basis for the development of medical and veterinary procedures. Various health disorders from different fields of red kangaroo medicine have been published. These include veterinary toxicology [3], oncology [4], urology [5], gastroenterology [6], ophthalmology [7], orthopedics [8], and parasitology [9]. In other species from the Macropodidea superfamily, for example, the Bennet’s wallaby, a heart pathology for hypertrophic cardiomyopathy was described [10]. Vascular pathological changes suggestive of hypertension were also described in western gray kangaroos (*Macropus puliginosus*) [11]. The description of a pattern of the arteries in the brain region is also important during brain surgeries since it enables planning and locating precise ligation of the vessels supplying blood to the brain [12]. This disease may lead to blockage of the arteries (e.g., in the brain area) by blood clots affecting cerebral vasculature [13]. Arterial blood flow is crucial for the development and proper functioning of every system in the body, so describing detailed anatomy of the arterial pattern of particular body regions is important for establishing new protocols for diagnostics and treatment by veterinary clinicians. Vasculature of the brain and the mechanisms that control this region’s blood pressure and temperature in kangaroos have been researched before, but no clear results have been described [14]. Such studies may also benefit from anatomical studies of specific regions of the body. Understanding the topographical relations of the blood vessels in the body may also help in understanding the physiological and pathophysiological mechanisms of different processes and disease development.

Anatomical studies on the different species of kangaroos are scarce. The general anatomy of the red kangaroo was described in 1897 by Windle [15]. Since then, anatomical techniques have increased significantly, and much more detailed structures can be described. Published anatomical research on kangaroos focused on the digestive system [16], penile anatomy [17], myology [18], and heart anatomy [19]. New reports in the field of detailed vascular anatomy can bring a lot of information that complements numerous studies on the evolution or biology of individual species.

The aim of this study was to describe the arterial blood supply to the brain and the remaining area of the derivatives of the internal carotid artery supply of the red kangaroo and perform an insightful comparative analysis of the anatomy of the arterial pattern of these regions described in other mammalian species.

## 2. Materials and Methods

### 2.1. Animals

This study was conducted on 18 adult red kangaroos (*Osphranter rufus*). Eight males and ten females were used for the purpose of this research. The age of the individuals was 2–6 years, and the body weight was 23–41 kg. The specimens were obtained from zoological gardens in Poland. In those zoos, the animals are kept in high-quality enclosures meeting species-specific requirements and are under constant veterinary care. They are given necessary prophylaxis treatment, e.g., antiparasitic medications. All animals included in the research for this article were euthanized (xylazine 10 mg/kg (intramuscular, i.m.), ketamine 90 mg/kg (i.m.), and pentobarbital 100 mg/kg (intravenous, i.v.)) for medical reasons other than head trauma, neurological disorders, or cardiovascular disease. None of the animals used in this study were euthanized for the purpose of this experiment. Prior to this study, all of the specimens were frozen.

### 2.2. Methods

In this study, a few different anatomical techniques were used in order to obtain high-quality research material and pictures. Both traditional anatomical techniques used in vascular anatomy research and a high-quality advanced imaging technique (cone-beam computed tomography, CBCT) were used. For each method, specimens of both genders were randomly assigned.

#### 2.2.1. Corrosion Casting

The first method was used in eight specimens. The cadavers were cut prior to the diaphragm, behind the seventh rib. The thoracic part of the descending aorta (*aorta thoracica*) was injected with a tinged solution of the chemo-setting acrylic material Duracryl^®^ Plus (SpofaDental, Jičín, Czech Republic) with a tinged solution of the chemo-setting acrylic material Duracryl^®^ Plus (SpofaDental, Jičín, Czech Republic). After the injection, when the acrylic material hardened, the specimens were submerged in washing powder solution (Persil, Germany) for the process of maceration. The temperature used for this process was 40 °C. The process lasted 30 days. The preparations acquired by this method were castings of the arterial vessels running from an aortic arch to the forelimb, thoracic cavity, neck, and head on a bone scaffold.

#### 2.2.2. Latex Preparations

The second method used in eight animals was injecting liquid LBS 3060 latex dyed red. The cadavers were cut prior to the diaphragm, and the material was introduced into the thoracic part of the descending aorta, and then the preparations were cured in 5% formaldehyde solution. After 20 days in the solution, the specimens were rinsed with tap water for 48 h to flush out the formaldehyde. Additional protection used in this method included wearing masks with high-quality filters and working in a preparation room with an efficient ventilation system. The system settings were 18 air changes per 1 h. The preparation of such specimens included butting the skull bone carefully, using an oscillating saw. This enabled soft tissue preparation inside the cranial cavity. The preparation of the vessels was performed by hand, using surgical instrumentation. Preparation began with removing skin from the whole analyzed area (head, neck, and part of the chest). The chest was opened by cutting the ribs over the sternum. After this, the muscle tissue was gently prepared to visualize injected arteries. The excess of connective tissue was removed in order to see the vascular pattern more clearly.

#### 2.2.3. CBCT Scanning

The third method used in this study was used in two cadavers of both genders. It was a CT angiography examination of the head and neck. CBCT (Fidex Animage, Pleasanton, CA, USA) was used for better imaging of small anatomical structures. The heads were fixed still on the tomograph’s table. Prior to the scan, the cadavers were cut prior to the diaphragm, and the thoracic part of the descending aorta was injected with contrast agent (barium sulphate; barium sulphuricum 1.0 g/mL, Medana, Sieradz, Poland). The scans were performed at the University Centre for Veterinary Medicine in Poznan, Poland, with scanning parameters of 110 kVp, 0.08 mAs per shot, 20.5 mAs (total mAs), and a reconstructed slice thickness of 0.3 mm. After the examination, the scan were studied and proper images were taken in FidexGUI (version 3.6.0, Animage, Pleasanton, CA, USA) with maximum intensity projection image reconstruction.

The names of the anatomical structures were standardized according to Nomina Anatomia Veterinaria [20].

All of the photographs taken during the study was made with a digital camera (Nikon D3200). The photographs were saved in JPEG format. GIMP v2.10.18 digital image editing software was used to process the photographs.

## 3. Results

Blood enters the aorta from the left ventricle of the heart. From the aortic arch, the brachiocephalic trunk (truncus brachiocephalicus) (Figure 1) and the left subclavian artery (arteria subclavia sinistra) branch off. The brachiocephalic trunk is a very short vessel. It is divided into the right subclavian artery and the bicarotic trunk (truncus bicaroticus).

One of the vessels into which the subclavian artery divides is the vertebral artery (arteria vertebralis). This vessel enters the transverse process openings (foramen processus transversus) of the cervical vertebrae. Next, the vertebral artery enters by the lateral vertebral opening (foramen vertebrale laterale) of the atlas into the vertebral canal (canalis vertebralis). Bilateral vertebral arteries unite and form the single basilar artery (arteria basilaris). The basilar artery enters the foramen magnum, runs along the bottom of the cranial cavity and rostrally to join the caudal communicating arteries (arteriae communicans caudales). The basilar artery has the same diameter (mean 0.85 mm) along the entire length of its course (Figure 2).

From the basilar artery, the caudal cerebellar arteries (arteriae cerebelli caudales) branch off (Figure 3). These are a single vessel on each side of the body. This vessel is arranged in the hemispheres of the cerebellum and supplies its caudal part. Moreover, from the basilar artery, several small arteries lead to the medulla oblongata.

More rostrally, the basilar artery joins the caudal communicating arteries. These arteries branch off from the internal carotid artery (arteria carotis interna) by a common trunk with the caudal cerebral artery (arteria cerebri caudalis). In three of the specimens, there was no common trunk for this vessel, and the caudal cerebral artery and the caudal communicating artery branched off independently from the internal carotid artery. Bilateral caudal communicating arteries merge with a single basilar artery, which makes the cerebral arterial circle sealed from the caudal side (Figure 4). From the caudal communicating artery, the rostral cerebellar artery (arteria cerebelli rostralis) branches off. This artery lies on the rostrolateral side of the cerebellum. Vessels located more rostrally branch off from the internal carotid artery.

The blood reaches the internal carotid artery from the bicarotic trunk as follows. The bicarotic trunk divides into two (right and left) common carotid arteries (arteriae carotes communes). These vessels are arranged below the cervical vertebrae close to the trachea and direct towards the head. Below the atlas, from the common carotid artery, the internal carotid artery branches off. In the initial part of this vessel, there is no carotid sinus (sinus caroticus). The internal carotid artery is a strong vessel (mean 1.4 mm) throughout its entire length (Figure 5). It enters the cranial cavity via the carotid canal and stacks on the base of the sphenoid bone. The first branch, located the most caudally, is the common trunk for the caudal cerebral artery and the caudal communicating artery. The caudal cerebral artery is a small-lumen (mean 0.7 mm) vessel that supplies the caudal part of the brain but also heads to the cerebellum from its rostral side. The second vessel which branches off from the internal carotid artery is the rostral cerebral artery (arteria cerebri rostralis) (mean 0.63 mm). From its initial part, the rostral choroidal artery (arteria choroidea rostralis) branches off. This vessel runs upwards between the brain’s temporal lobe and the cerebral crus. Afterward, from the rostral cerebral artery, the middle cerebral artery (arteria cerebri media) branches off. This vessel’s lumen is bigger (mean 1.08 mm). It runs dorsally and surrounds the brain’s piriform lobe from its rostral side and branches into several branches.

The middle cerebral artery has four main branches. From the emergence of the middle cerebral artery, two vessels branch off: the frontal and orbital branches. Altogether, those vessels were observed to branch off united as a common trunk. The frontal branch further gives two smaller vessels once again: the ventral frontal branch and the dorsal frontal branch. The caudal olfactory artery branches off at the level of the common trunk beginning point. Next, in the caudal direction, the temporal branches originate (divided secondarily into the middle and ventral temporal branches). At the end, the middle cerebral artery is divided into the parietal branch (divided secondarily into the rostral and caudal parietal branches) and the dorsal temporal branch. The dorsal temporal branch is the vessel described to have the largest diameter (mean, 0.6 mm). More rostrally to the middle cerebral artery, the rostral communicating artery (arteria communicans rostralis) branches off. This vessel is present in 14 analyzed specimens. It connects bilateral rostral cerebral arteries as a small-lumen vessel. In a few of the specimens, this artery is in the form of several small vessels. The presence of this vessel makes the cerebral arterial circle closed from the anterior side. The rostral cerebral artery was observed to run in the rostral direction and then change, lying in the longitudinal fissure of the brain.

Between the olfactory bulbs, the internal ethmoidal artery (arteria ethmoidalis interna) branches off, heading to the cribriform plate of the ethmoid bone. From this vessel, tiny branches supplying the medial surface of the olfactory bulbs branch off. The shape of the cerebral arterial circle is specific. In fact, it is difficult to name this structure a circle. The cerebral arterial circle is similar in shape to a pentagon, and the caudal side is in the form of a semicircle (Figure 4). Lateral sides are built of straight vessels. From the caudal side, the shape is closed by arcing caudal communicating arteries and from the anterior side by the rostral cerebral arteries arranged convergently in the shape of a triangle. These vessels are connected by the rostral communicating arteries. When the rostral cerebral artery branches off the internal carotid artery, this last vessel passes through the optic canal as the internal ophthalmic artery (arteria ophthalmica interna). This is the only source of blood for the orbit. Moreover, one branch from this artery passes through the ethmoid foramen as the external ethmoidal artery (arteria ethmoidalis externa) to vascularize the ethmoid labyrinth.

## 4. Discussion

Windle and Parsons described the general vascular anatomy of the red kangaroo in 1897. The authors only mentioned the topography of the common carotid artery bifurcation and mentioned similarities in the anatomy of the subclavian artery to previously described related species from the Macropodidea superfamily—Petrogale [14]. Since then, no detailed description of the arterial circulation of the head in this species has been described. The vascular pattern of the main vessels from the aortic arch in a red kangaroo is the same as that described in a rabbit [21] and a domestic pig [22]. In these species, the left subclavian artery and the brachiocephalic trunk branch off from the aortic arch. After the right subclavian artery branches off, the bicarotic trunk is observed. The vertebral artery branches off from the subclavian artery [21]. The most similar pattern of vessels was described in the cat [23], dog [24], red squirrel [25], and ground squirrel [26]. In these species, the bicarotic trunk is absent. The left common carotid, right common carotid, and right subclavian artery branch off from the brachiocephalic trunk. The vasculature pattern present in a donkey [27], Siberian deer [28], pampas deer, and brown brocket deer [29] branches off from the brachiocephalic trunk. Moreover, in the brown brocket deer, there is no bicarotic trunk [29], and in the Siberian deer, the right and left vertebral arteries branch off from the right and left costocervical trunks (truncus costocervicalis), which branch off from the brachiocervical trunk, not from the subclavian artery [28]. A different vascular system has been described in the axis deer. In this case, the right/left common carotid artery branches off from the right/left subclavian artery [29]. More vessels branch off in the rat [21], porcupine [30], mole-rat [31], giant armadillo [32], and human [33]. In these species, the brachiocephalic trunk, left common carotid, and left subclavian artery branch off. The brachiocephalic trunk is divided into the right subclavian and right common carotid artery. The well-developed internal carotid artery is observed in the red-necked wallaby [34] but also in representatives of the lagomorphs [35,36], Caniformia [22,37,38,39,40], odd-toed ungulates [22,41], and Cameliformes [42,43,44] as well as some rodents like agouti [45], Mongolian gerbil [46], European beaver [47], Egyptian spiny mouse [48], and rat [49]. The shape of the cerebral arterial circle in the red kangaroo is like that in the red-necked wallaby [34]. In other groups of animals, shape varies greatly. The cerebral arterial circle is heart-shaped in representatives of the ruminants, such as the common wildebeest, Antilopinae, Tragelaphus, Taurotragus, or Boselaphus [50,51,52,53]. The triangular shape is noted in the giraffe [54], and in tapir, camels, and cats, it is like the numeral “8” [41,42,55]. In equines, it is a rectangular shape [41]. In rodents and panther, it is an ellipse [26,45,55,56,57,58,59,60,61]. It is described as an hourglass in the lagomorph [35,36]. The rostral communicating artery was observed in most of the analyzed red kangaroo preparations. It was not, however, present in all of the specimens. Fenrich (2021) believes that this artery is involved in forebrain dehydration detection [62]. He suggests that animals possessing this artery are advantaged in detecting dehydration. The loss of water effects a reduction in extracellular fluid, which leads to decreased tissue perfusion, resulting in a shift to anaerobic metabolism in the peripheral tissues [63]. The brain has a very limited capacity for anaerobic metabolism and is therefore a privileged organ during dehydration [64]. Anaerobic metabolism in the brain is avoided and delayed with adequate blood supply [64,65]. Having sensors of dehydration in the vicinity of neurons involved in the relevant effector homeostatic mechanisms is rather beneficial for the functioning of this process.

However, numerous works indicate that the occurrence of this vessel is an individual rather than a species issue. In various studies of different species, this vessel was present in some of the analyzed specimens, and in other individuals, it was not. This vessel was observed in 78% of red-necked wallabies [34], 75% of American bison, 84% of domestic cattle, and 80% of European bison [66]. In studies conducted by König on domestic cattle, in most cases, no rostral communicating artery was observed [67]. Research carried out by Kuchinka on the chinchilla proves that this vessel was present in 23% of specimens [61]. Similar observations were noted by Reckezegiel et al. on the capybara [68]. The absence of this vessel as a species trait was noted in the guinea pig [60,69] and European ground squirrel [70].

The rostral communicating artery in the red kangaroo was noted in most specimens as a single vessel and in some as several small vessels. This is a single vessel in, e.g., porcupine, rat, and mouse [57,71,72]. This artery was noted as a network of small vessels in cattle, sheep, goat, and Eurasian elk [73,74]. Moreover, in some cases of Eurasian elk, this artery was asymmetric, branching off from the rostral cerebral artery before the middle cerebral artery branched off [74].

The middle cerebral artery supplies blood to most of the cerebrum. In the red kangaroo, the middle cerebral artery supplies the same areas of the brain as in the mammalian species studied so far. The branching off of vessels in different species is subject to variability. In some representatives of Carnivora, such as the otter or silver fox, dorsal and ventral frontal branches branch off independently, not by creating a common trunk [40,75]. The temporal branches are the same, branching off from the main trunk independently. Finally, the main vessel is divided into parietal branches [75]. The vascular pattern in the red kangaroo is more similar to the one described in the polar fox. Three main branches of the middle cerebral artery were observed. The rostral middle cerebral artery is divided into the orbital, ventral frontal, and dorsal frontal branches. The second one is the dorsal middle cerebral artery. It is divided into the rostral and caudal parietal branches. The last one is the caudal middle cerebral artery, from which emerge the dorsal, middle, and ventral temporal branches [76]. In the red kangaroo, the internal ophthalmic artery is a well-developed vessel. It is a continuation of the internal carotid artery. The well-developed internal ophthalmic artery was noted in the red-necked wallaby [34], guinea pig [60], and Patagonian mara [77]. There is a fundamental difference between Macropodidea superfamily members and these rodents in this vascular area. In the wallaby and the kangaroo, there is no connection between the internal ophthalmic artery and the maxillary artery (via the external ophthalmic artery). The internal ophthalmic artery evolved into the external ophthalmic artery in the guinea pig and Patagonian mara. This last vessel is connected to the maxillary artery. In the red kangaroo, the well-developed external ethmoidal artery is noted. From the even-toed ungulates and carnivores, most species have this artery as a primary source of blood to the ethmoid labyrinth. Such a pattern was described in representatives of Giraffidae, Bovidae, Cervidae, Camelidae, Hippopotamidae, Canidae, Mustelidae, Procyonidae, and Phocidae. Only in Suidae, Tayassuidae, and Felidae representatives was the main source of blood the internal ophthalmic artery [78]. In most specimens, the caudal cerebral artery branched off from the internal carotid artery by a common trunk with the caudal communicating artery. In the red-necked wallaby, in most cases, the caudal cerebral artery branches off independently from the internal carotid artery [34]. The caudal cerebral artery is a single vessel (on each side of the body) like in most species. The double vessel was noted in the European roe deer [79] and Patagonian mara [77]. The basilar artery is a strong vessel with the same diameter over its entire length. In contrast to the experiment performed by Balwid and Bell [80,81,82] on cattle and sheep, where the diameter of the basilar artery was measured to assess the involvement of this vessel in the cerebral and cerebellar circulation, in red kangaroo, the diameter of the basilar artery does not decrease caudally. This decrease in the diameter observed in cattle and sheep was an anatomical indicator for the support of the theory that the vertebral arteries do not supply the basilar artery. The constant diameter of this vessel in red kangaroo may contribute to the theory that the vertebral arteries contribute to the blood supply of the caudal part of the encephalon.

In the described species of rodents—the ground squirrel, chinchilla, porcupine, pacarana, capybara, red squirrel, and nutria, the only source supplying blood to the brain is the basilar artery originating from the vertebral arteries [56,57,58,61,68,70,83]. In those species, the internal carotid artery was not observed.

Another described arterial pattern of this region was described in degu. In this species, along with a basilar artery, a narrow internal carotid artery supplying a small amount of blood to the arterial circle of the brain was described [59].

The next pattern of arterial model which includes red kangaroo was described in red-necked wallaby, agouti, mouse, gerbil, hamster, and vole. In these cases, both the basilar artery and internal carotid artery supply blood to the brain [34,45,46,48,84].

In the Patagonian mara and the guinea pig, the rostral cerebral artery is joined by the internal ophthalmic artery along with the strong basilar artery and less developed internal carotid artery [60,77].

According to De Vriese (1905), there are three types of arterial blood supply to the brain. In type I, the brain is supplied exclusively by the carotid system. In type II, both the carotid and the vertebral–basilar system supply the brain evenly or with a predominance of one of them. In type III, the brain irrigation depends almost exclusively on the vertebral–basilar system [85]. Based on the results of this study, the arterial vascularization of the brain in red kangaroo is classified as type II. The same type was previously described in agoutis, cats, dogs, horses, opossums, rabbits, coatis, and pigs [45,86,87,88,89,90,91,92].

## 5. Conclusions

This study provided a detailed description of the vascular anatomy of the brain and nearby regions of the red kangaroo (*Osphranter rufus*). Different methods were used to visualize arteries and enable topographical descriptions of the vessels’ courses. The described finding has been discussed and compared with research on different species of mammals. The arteries at the base of the encephalon in the red kangaroo are arranged similarly to the red-necked wallaby. The primary source of blood to the encephalon is the internal carotid artery. The secondary source is the vertebral artery. The internal carotid artery supplies the brain and partially the cerebellum. This vessel vascularizes also the orbit and the ethmoid labyrinth. The cerebellum is vascularized in addition from the vertebral artery. The results of this study provide the first description of vascularization of this area in the red kangaroo but may also help in future research on physiology, pathophysiology, or different branches of veterinary medicine.

## Figures and Tables

**Figure 1 animals-13-02744-f001:**
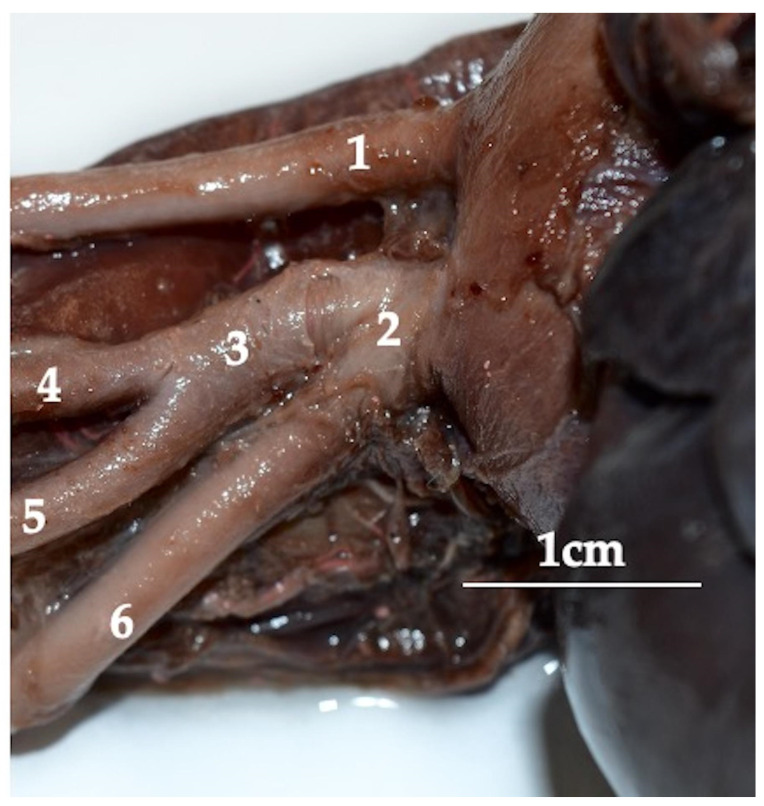
The arteries branching off from an aortic arch of a red kangaroo. Latex preparation. 1—left subclavian artery (arteria subclavia sinistra); 2—brachiocephalic trunk (truncus brachiocephalicus); 3—bicarotic trunk (truncus bicaroticus); 4—left common carotid artery (arteria carotis communis sinistra); 5—right common carotid artery (arteria carotis communis dextra); 6—right subclavian artery (arteria subclavia dextra).

**Figure 2 animals-13-02744-f002:**
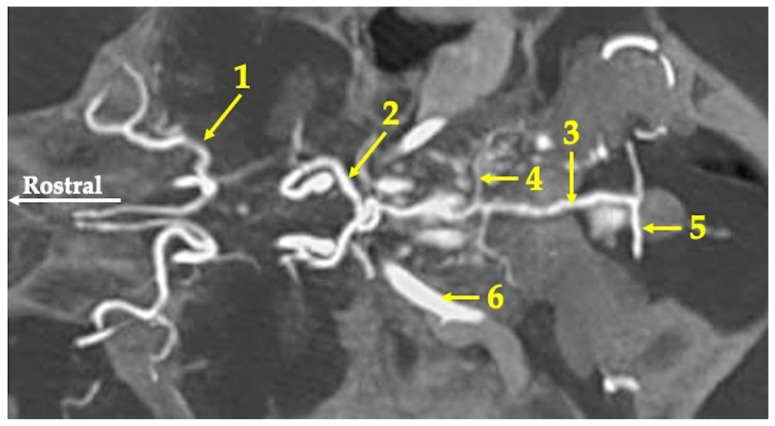
Maximum intensity projection of the agioCT scan of the head of the red kangaroo. Axial plane at the level of the bottom of the cranial cavity. 1—middle cerebral artery (arteria cerebri media); 2—caudal communicating artery (arteria communicans caudalis); 3—basilar artery (arteria basillaris); 4—caudal cerebral artery (arteria cerebri caudalis); 5—vertebral artery (arteria vertebralis); 6—internal carotid artery (arteria carotica interna).

**Figure 3 animals-13-02744-f003:**
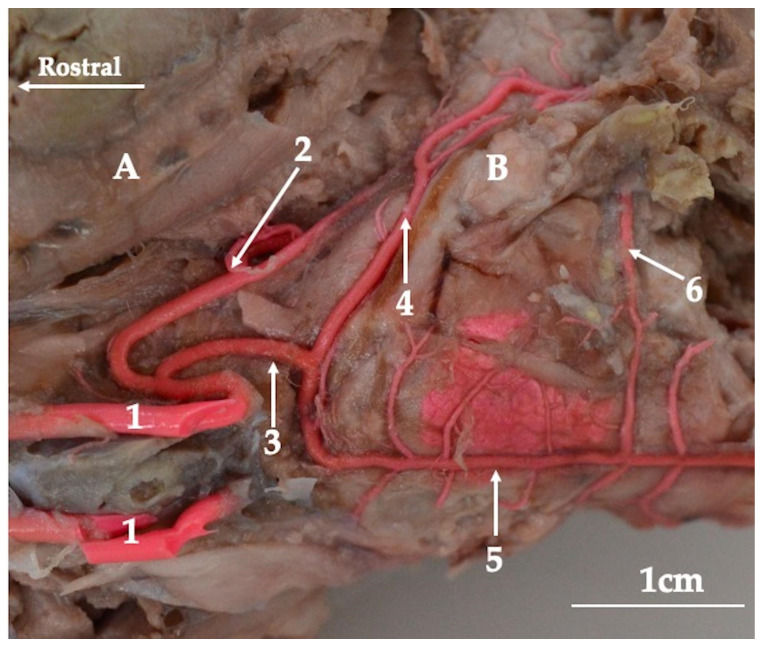
The arteries at the base of the encephalon of the red kangaroo. Latex preparation. Ventrolateral view. A—the brain (cerebrum); B—the cerebellum (cerebellum); 1—internal carotid artery (arteria carotica interna); 2—caudal cerebral artery (arteria cerebri caudalis); 3—caudal communicating artery (arteria communicans caudalis); 4—rostral cerebellar artery (arteria cerebelli rostralis); 5—basilar artery (arteria basillaris); 6—caudal cerebellar artery (arteria cerebelli caudalis).

**Figure 4 animals-13-02744-f004:**
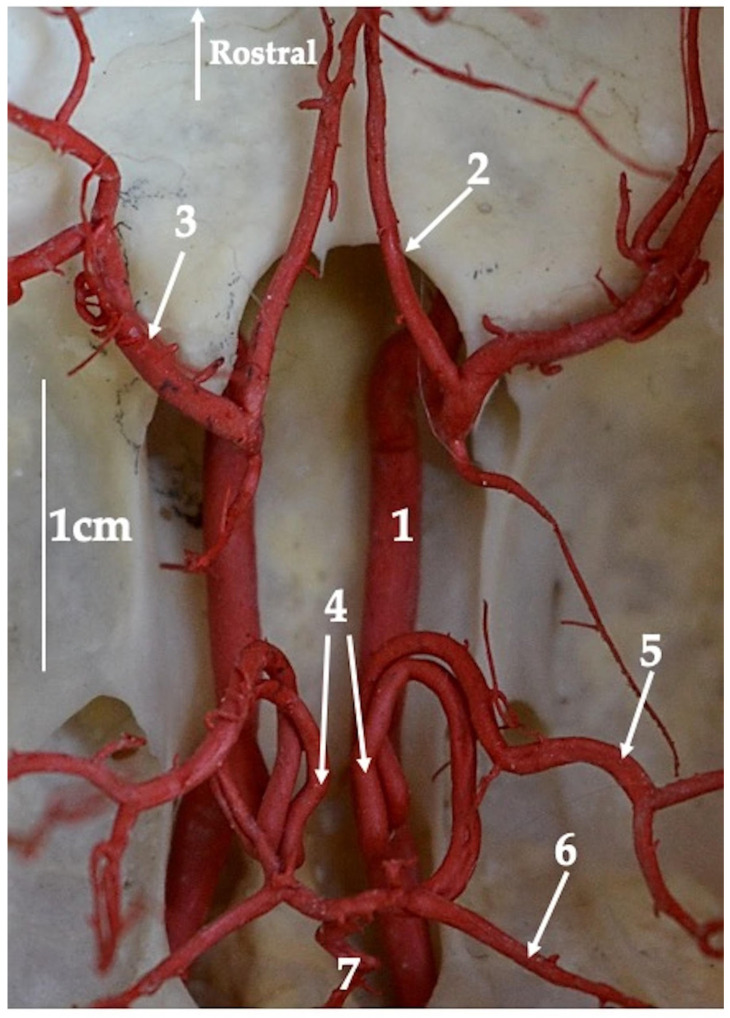
Dorsal view of the arteries forming arterial circle of the brain at the base of the cranial cavity of the red kangaroo. Corrosion cast. 1—internal carotid artery (arteria carotica interna); 2—rostral cerebral artery (arteria cerebri rostralis); 3—middle cerebral artery (arteria cerebri media); 4—caudal communicating artery (arteria communicans caudalis); 5—caudal cerebral artery (arteria cerebri caudalis); 6—rostral cerebellar artery (arteria cerebelli rostralis); 7—basilar artery (arteria basillaris).

**Figure 5 animals-13-02744-f005:**
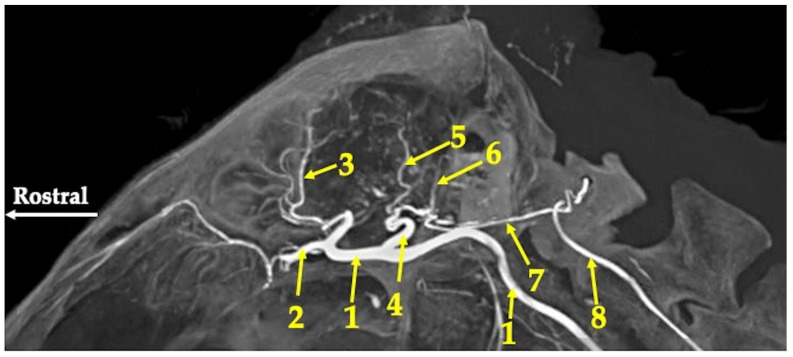
Maximum intensity projection of the agioCT scan of the head of the red kangaroo. Saggital plane. 1—internal carotid artery (arteria carotica interna); 2—internal ophthalmic artery (arteria ophthalmica interna); 3—rostral cerebral artery (arteria cerebri rostralis); 4—caudal communicating artery (arteria communicans caudalis); 5—caudal cerebral artery (arteria cerebri caudalis); 6—rostral cerebellar artery (arteria cerebelli rostralis); 7—basilar artery (arteria basillaris); 8—vertebral artery (arteria vertebralis).

## Data Availability

The data presented in this study are available on request from the corresponding author.

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
