# Peer review of "Cerebral Vascularization and the Remaining Area Supply of the Internal Carotid Artery Derivatives of the Red Kangaroo (Osphranter rufus)"

_animals, 2023, doi:10.3390/ani13172744_

Round 1

Reviewer 1 Report

The work is relevant and demonstrates merit. However, the authors should make some revisions. Enhance the description of the involvement of the vertebral artery and cerebral blood supply. Review the course and point of entry of the internal carotid artery into the cranial cavity. Thoroughly examine the arrangement and contribution of the internal ethmoidal artery to cerebral blood supply. Some descriptions regarding the role of the carotid-vertebral system in different species are confusing and warrant a new literature review. The origin of the common carotid artery is not, in my view, significant, even when considering the diversity found among different species. Hence, the authors should indeed emphasize which vessels collaborated and could have collaborated, according to the different species investigated. Particularly, the constitution of the carotid-basilar system. Systematically, the authors should reconsider the importance of the rostral communicating artery, in comparison with the formation of the arteries at the base of the brain, and expand the literature review on this topic. Reassess the significance of the internal ophthalmic artery, its origin, and contribution, also highlighting the involvement of the maxillary artery. As this concerns a system, it would be beneficial for the authors to evaluate the foundational works regarding the different classifications proposed by De Vriese (1905).

Author Response

Ms. Bojana Nadaždin

Assistant Editor - Animals

Re: Your submission to Animals - Manuscript ID: animals-2567576

We are pleased to resubmit to you our manuscript (Manuscript ID: animals-2567576) entitled “Cerebral vascularization and the remaining area supply of the internal carotid artery derivatives of the red kangaroo (Osphranter rufus)". Thank you for your decision letter and your time regarding our revision.

We would like to thank the Editor and the Reviewers for all the valuable comments and suggestions, which have helped us to improve our manuscript.

We have introduced changes indicated by the Reviewers’ comments and concerns. The manuscript has been linguistically corrected.

Sincerely yours

Jakub J. Ruszkowski

jakub.ruszkowski@up.poznan.pl

Reviewer 1

The work is relevant and demonstrates merit. However, the authors should make some revisions.

Enhance the description of the involvement of the vertebral artery and cerebral blood supply.

The description of the blood supply from the vertebral artery enhanced in the discussion sections. Lines – 349-257

Review the course and point of entry of the internal carotid artery into the cranial cavity.

The information added – line 204.

Thoroughly examine the arrangement and contribution of the internal ethmoidal artery to cerebral blood supply.

The information added – line 241-242.

Some descriptions regarding the role of the carotid-vertebral system in different species are confusing and warrant a new literature review.

The origin of the common carotid artery is not, in my view, significant, even when considering the diversity found among different species. Hence, the authors should indeed emphasize which vessels collaborated and could have collaborated, according to the different species investigated. Particularly, the constitution of the carotid-basilar system.

The comparative analysis of vascularization of this regions has been added to the discussion section. Lines – 358-370.

Systematically, the authors should reconsider the importance of the rostral communicating artery, in comparison with the formation of the arteries at the base of the brain, and expand the literature review on this topic.

Done as requested - lines 311-316.

Reassess the significance of the internal ophthalmic artery, its origin, and contribution, also highlighting the involvement of the maxillary artery. As this concerns a system, it would be beneficial for the authors to evaluate the foundational works regarding the different classifications proposed by De Vriese (1905).

Done as requested - lines 371-377.

Reviewer 2 Report

I found this article very interesting. While I do have some suggestions and questions (see below), I believe this work is an important contribution to wildlife medicine.

1)     I advice you to speak in the introduction section of the actual state of art of surgery in this region, to underline why your study is so important for future surgery

2)     In lines #50-52, the sentence („Hearth pathology – hypertrophic cardiomyopathy was described in a related species – Bennett’s wallaby [10]”) seems to be incomplete.

3)     In line #55 you write about blot knots in relation with the hypertrophic cardiomyopathy. Is there any blood analysis or other study in this (or related) species which indicates probability of such problems?

4)     In line #65, when referring to previous anatomical descriptions, the extent of those should be mentioned. Which of the arteries detailed  in the current study were described in the previous ones? Also, similarities/differences with the previous studies should be mentioned in the discussion part.

5)      Line #78 – what is the average bodyweight and age of the subjects?

6)       Line # 94 - „The cadavers were cut prior to the 94 diaphragm” – exactly where? The diaphragm has a concave curvature, and “prior” is not an anatomical direction, it can mean anything. Nr of the thoracic vertebrae should be you used.

7)      “aorta descendens pars thoracica” – this is not a standard anatomical name from your reference (NAV 6th), “aorta thoracica” should be used instead.

8)      Section 2.2 should be divided into 3 subsections according to the different methods used. Were there any differences in the results based on the methods? Is it possible that an inconstant vessel was simply not approachable with a given method, hence the inconsistency in its presence?

9)      Line #163: The spinal cord’s anatomical name is medulla spinalis, not medulla oblongata, which one are you talking about?

10)   Line #193: you mention the lack of carotid sinus which is fascinating. Is this observation unique or other species mentioned in this section also have this feature? Also, is there any trace of the corresponding  baro-, and chemoreceptors in that area? Did you happen to see the Herring nerve approaching the corresponding section anyway? If not, where the receptor center could be in this species?

11)   you mention that some vessels are larger or smaller than others – can you indicated any approximate diameters?

12)   Line #221: “In this species, the dorsal temporal branch was the vessel described to have the largest diameter.” Where was this described? Citation? How large is this?

13)   Some confusion about the external ophthalmic artery. In line 238 the a. ophtalmica interna is explained to be the only vessel for the orbit. In line #301 this is reassured and the definition is expanded for all kangaroos (although without citation!). But in line #304: “In the red kangaroo noted, the well-developed external ophthalmic artery.” – this sentence is not comprehensive grammar wise, and contradicts the previous statement, please clarify.

14)   IMAGES:

a.        write “ 1 cm” onto the scale bar of the images

b.      use the same font as that of the journal, an be consistent with the font sizes

c.       Instead of writing both “Cranial” and “Caudal” an arrow with one expression should be used.

d.      the direction “cranial” should not be used in this are at all. The authors themselves use the “rostral” direction throughout the article except for the images.

e.      the abbreviation “MIP” is defined once in the article and twice in the images (Fig 2 & 5). They are not used later in the article or in any of these picture descriptions, so the abbreviation is a surplus.

Author Response

Ms. Bojana Nadaždin

Assistant Editor - Animals

Re: Your submission to Animals - Manuscript ID: animals-2567576

We are pleased to resubmit to you our manuscript (Manuscript ID: animals-2567576) entitled “Cerebral vascularization and the remaining area supply of the internal carotid artery derivatives of the red kangaroo (Osphranter rufus)". Thank you for your decision letter and your time regarding our revision.

We would like to thank the Editor and the Reviewers for all the valuable comments and suggestions, which have helped us to improve our manuscript.

We have introduced changes indicated by the Reviewers’ comments and concerns. The manuscript has been linguistically corrected.

Sincerely yours

Jakub J. Ruszkowski

jakub.ruszkowski@up.poznan.pl

Reviewer 2

I found this article very interesting. While I do have some suggestions and questions (see below), I believe this work is an important contribution to wildlife medicine.

1)     I advice you to speak in the introduction section of the actual state of art of surgery in this region, to underline why your study is so important for future surgery

Sentene about importance of the regions anatomy for surgery added. Lines – 57-59.

2)     In lines #50-52, the sentence („Hearth pathology – hypertrophic cardiomyopathy was described in a related species – Bennett’s wallaby [10]”) seems to be incomplete.

Sentence rephrased – 53-55.

3)     In line #55 you write about blot knots in relation with the hypertrophic cardiomyopathy. Is there any blood analysis or other study in this (or related) species which indicates probability of such problems?

There is one study on hypertrophic cardiomyopathy in related species (Bennet’s wallaby). The study is cited as the reference number 10.

4)     In line #65, when referring to previous anatomical descriptions, the extent of those should be mentioned. Which of the arteries detailed  in the current study were described in the previous ones? Also, similarities/differences with the previous studies should be mentioned in the discussion part.

Done as requested - lines 255-260.

5)      Line #78 – what is the average bodyweight and age of the subjects?

Done as requested - lines 84-85.

6)       Line # 94 - „The cadavers were cut prior to the 94 diaphragm” – exactly where? The diaphragm has a concave curvature, and “prior” is not an anatomical direction, it can mean anything. Nr of the thoracic vertebrae should be you used.

Done as requested – line 103.

7)      “aorta descendens pars thoracica” – this is not a standard anatomical name from your reference (NAV 6th), “aorta thoracica” should be used instead.

Done as requested – lines 103-104.

8)      Section 2.2 should be divided into 3 subsections according to the different methods used. Were there any differences in the results based on the methods? Is it possible that an inconstant vessel was simply not approachable with a given method, hence the inconsistency in its presence?

Done as requested.

The results of those methods were consistent. We decided to use those methods to enhance the ways to visualize the vessels. Latex preparations are better for visualizing vessels lying between the soft tissue and corrosion those perforating bones. CBCT scans help to asses the topography of vessels in the head in 3D dimensions.

9)      Line #163: The spinal cord’s anatomical name is medulla spinalis, not medulla oblongata, which one are you talking about?

Corrected. Line 173.

10)   Line #193: you mention the lack of carotid sinus which is fascinating. Is this observation unique or other species mentioned in this section also have this feature? Also, is there any trace of the corresponding  baro-, and chemoreceptors in that area? Did you happen to see the Herring nerve approaching the corresponding section anyway? If not, where the receptor center could be in this species?

We have no knowledge about the location of the baro- and chemoreceptors in this area. During the preparation, we did not pay close attention to the nerves related to the prepared area, since the goal was to obtain a view of the arterial vessels as clearly as possible.

11)   you mention that some vessels are larger or smaller than others – can you indicated any approximate diameters?

The mean diamateres of more important vessels were added to the text.

12)   Line #221: “In this species, the dorsal temporal branch was the vessel described to have the largest diameter.” Where was this described? Citation? How large is this?

Sentence rephrased. It is our result. The sentence was incorrect English.

13)   Some confusion about the external ophthalmic artery. In line 238 the a. ophtalmica interna is explained to be the only vessel for the orbit. In line #301 this is reassured and the definition is expanded for all kangaroos (although without citation!). But in line #304: “In the red kangaroo noted, the well-developed external ophthalmic artery.” – this sentence is not comprehensive grammar wise, and contradicts the previous statement, please clarify.

Text was rephrased. Sorry for the mistake.

14)   IMAGES:

  1. write “ 1 cm” onto the scale bar of the images

Done as requested.

  1. use the same font as that of the journal, an be consistent with the font sizes

Done as requested.

  1. Instead of writing both “Cranial” and “Caudal” an arrow with one expression should be used.

Done as requested.

  1. the direction “cranial” should not be used in this are at all. The authors themselves use the “rostral” direction throughout the article except for the images.

Done as requested.

  1. the abbreviation “MIP” is defined once in the article and twice in the images (Fig 2 & 5). They are not used later in the article or in any of these picture descriptions, so the abbreviation is a surplus.

Abbreviation deleted.

Reviewer 3 Report

This study concerns the arterial supply of the red kangaroo brain. Kangaroos are unique mammals, so it is appropriate to undertake research into their morphology. To date, no studies have been conducted on the vascularisation of this animal species. The vascularisation of the brain also has an impact on thermoregulation.  The aim of the study is therefore fully justified.

The research material is sufficient and allows concrete conclusions to be drawn.

The research methodology is a combination of traditional anatomical methods, i.e. preparation and intravascular injection, and computed tomography. I believe that this is as appropriate as possible. It is unfortunate that the CT scan was only performed on two specimens.

In the discussion, the authors compared the results with the available literature.

The figures are clear and provide a good illustration of the study.

The literature is appropriately selected and entirely sufficient.The paper is an interesting contribution to the anatomical knowledge of this animal species.

- Please specify at what time after death the CT scan was performed, this is important due to the post mortem swelling of the brain which obliterates and normal structures.

[97] eliminate the dot in between: (SpofaDental, Jičín, Czech Republic). with a tinged solution

Author Response

Ms. Bojana Nadaždin

Assistant Editor - Animals

Re: Your submission to Animals - Manuscript ID: animals-2567576

We are pleased to resubmit to you our manuscript (Manuscript ID: animals-2567576) entitled “Cerebral vascularization and the remaining area supply of the internal carotid artery derivatives of the red kangaroo (Osphranter rufus)". Thank you for your decision letter and your time regarding our revision.

We would like to thank the Editor and the Reviewers for all the valuable comments and suggestions, which have helped us to improve our manuscript.

We have introduced changes indicated by the Reviewers’ comments and concerns. The manuscript has been linguistically corrected.

Sincerely yours

Jakub J. Ruszkowski

jakub.ruszkowski@up.poznan.pl

Reviewer 3

This study concerns the arterial supply of the red kangaroo brain. Kangaroos are unique mammals, so it is appropriate to undertake research into their morphology. To date, no studies have been conducted on the vascularisation of this animal species. The vascularisation of the brain also has an impact on thermoregulation.  The aim of the study is therefore fully justified.

The research material is sufficient and allows concrete conclusions to be drawn.

The research methodology is a combination of traditional anatomical methods, i.e. preparation and intravascular injection, and computed tomography. I believe that this is as appropriate as possible. It is unfortunate that the CT scan was only performed on two specimens.

In the discussion, the authors compared the results with the available literature.

The figures are clear and provide a good illustration of the study.

The literature is appropriately selected and entirely sufficient.The paper is an interesting contribution to the anatomical knowledge of this animal species.

- Please specify at what time after death the CT scan was performed, this is important due to the post mortem swelling of the brain which obliterates and normal structures.

 The animals were frozen maximum 60 minutes after death.  The animals were defrozen in the water with the 30 degrees Celsius temperature.

[97] eliminate the dot in between: (SpofaDental, Jičín, Czech Republic). with a tinged solution

Done as requested - line 106.

Round 2

Reviewer 1 Report

accept